# Structural basis of RNA polymerase II transcription on the chromatosome containing linker histone H1

Rina Hirano [1,3], Haruhiko Ehara [2,3], Tomoya Kujirai[1,2], Tamami Uejima[2], Yoshimasa Takizawa[1], Shun-ichi Sekine [2] ✉ & Hitoshi Kurumizaka [1,2] ✉

In chromatin, linker histone H1 binds to nucleosomes, forming chromatosomes, and changes the transcription status. However, the mechanism by which RNA polymerase II (RNAPII) transcribes the DNA in the chromatosome has remained enigmatic. Here we report the cryo-electron microscopy (cryo-EM) structures of transcribing RNAPII-chromatosome complexes (forms I and II), in which RNAPII is paused at the entry linker DNA region of the chromatosome due to H1 binding. In the form I complex, the H1 bound to the nucleosome restricts the linker DNA orientation, and the exit linker DNA is captured by the RNAPII DNA binding cleft. In the form II complex, the RNAPII progresses a few bases ahead by releasing the exit linker DNA from the RNAPII cleft, and directly clashes with the H1 bound to the nucleosome. The transcription elongation factor Spt4/5 masks the RNAPII DNA binding region, and drastically reduces the H1-mediated RNAPII pausing.

Chromatin compacts genomic DNA in eukaryotes, and the nucleosome is an elemental architecture. The nucleosome is composed of linker DNAs and the nucleosome core particle (NCP), which folds an approximately 150 base-pair DNA segment into a histone octamer, comprising the four core histones H2A, H2B, H3, and H4. In the NCP, the DNA is symmetrically wrapped 1.7-turns around the histone octamer, and the central region of the nucleosomal DNA is located on the dyad axis[1]. NCPs are connected by linker DNA segments, and appear as a beads-on-a-string fiber[2].

Linker histones, such as H1, are major chromatin components that specifically bind to the nucleosomal DNA forming the chromatosome[3], and promote further compaction of chromatin[4–7]. H1 consists of three domains, the N-terminal disordered region (~35 residues), the central globular domain (~80 residues), and the C-terminal disordered region (~100 residues), and in the chromatosome, these domains mainly bind to a linker DNA, the DNA around the dyad axis, and another linker DNA, respectively[8–11]. This tripartite nucleosome binding by H1 is termed "on-dyad" binding, and it compacts the nucleosome architecture by restricting the linker DNA orientation and flexibility[12–14]. H1 also binds

the nucleosome in an "off-dyad" mode, in which the central globular domain mainly binds to a linker DNA[14–16].

H1 is the most abundant nucleosome binding protein, and exists in a 50–130% amount relative to nucleosomes[17]. H1 binding to the nucleosome induces structural changes in chromatin, thus regulating transcription processes catalyzed by RNA polymerase II (RNAPII)[4,5]. For instance, a biochemical study with *Drosophila* egg extracts indicated that H1 binding to reconstituted chromatin represses RNAPII transcription[18]. A mouse knock-out study revealed that embryos with an H1 ratio reduced to approximately 50% cease development at mid-gestation[19]. H1 reportedly functions in gene regulation through chromatin compaction and 3D genome organization[20]. Consistently, acute depletion of H1 leads to perturbations of gene expression in the constitutive heterochromatin in mouse embryonic stem cells[21]. H1 alleles have been identified as genes with driver mutations in lymphoma, revealing that H1 is a bona fide tumor suppressor[22]. These facts indicate that H1 plays pivotal roles in development, differentiation, and tumorigenesis by appropriately reducing transcription efficiency according to the cellular status.

[1]Laboratory of Chromatin Structure and Function, Institute for Quantitative Biosciences, The University of Tokyo, 1-1-1 Yayoi, Bunkyo-ku, Tokyo 113-0032, Japan. [2]RIKEN Center for Biosystems Dynamics Research, 1-7-22 Suehiro-cho, Tsurumi-ku, Yokohama 230-0045, Japan. [3]These authors contributed equally: Rina Hirano, Haruhiko Ehara. ✉e-mail: shunichi.sekine@riken.jp; kurumizaka@iqb.u-tokyo.ac.jp

Eleven non-allelic human H1 isoforms have been described, and they may have specific functions in altering the cellular transcription status, primarily by repressing transcription by RNAPII[5]. H1.2 is the major somatic linker histone. Genome-wide studies revealed that the genomic H1.2 binding loci negatively correlate with the RNAPII transcription activation, suggesting the repressor function of H1.2[23–25]. In contrast, genetic and biochemical studies demonstrated that H1.2 potentially upregulates gene expression[26–30]. Therefore, H1.2 may function to both repress and activate RNAPII transcription[5], probably depending on cell types and genomic loci. These different outcomes may be the results of complex transcription regulation at the levels of transcription initiation and elongation, but the direct effects of H1 on transcription are not well understood.

Linker histones are associated with the gene-body regions of actively transcribed genes[31,32]. H1 causes RNAPII pausing at the 3′ splice site, and regulates alternative splicing[32]. These observations suggested the specific involvement of linker histones in the regulation of transcription elongation, but the molecular details have remained elusive.

In the present study, we investigate the basic properties of the chromatosome in transcription elongation by performing in vitro transcription experiments and cryo-EM analyses, which reveal how H1 accomplishes nucleosome repression in transcription elongation.

## Results

### Linker histone H1 pauses RNAPII in front of the chromatosome

We first performed a nucleosome transcription assay in the presence of a linker histone, H1.2 (Fig. 1a, b and Supplementary Figs. 1, 2a–d). The nucleosome was reconstituted with a 261 base-pair (bp) DNA fragment and histones H2A, H2B, H3, and H4 (Supplementary Fig. 2c, d). The resulting nucleosome included the 92 bp entry linker DNA and the 24 bp exit linker DNA (Fig. 1b and Supplementary Fig. 1a). The entry linker DNA contained a 9-base mismatched region to form a transcription bubble with a primer RNA. RNAPII from the yeast *Komagataella pastoris* was loaded onto the bubble, and the transcription reaction was conducted in the presence of the transcription elongation factor TFIIS (Supplementary Figs. 1b, 2b).

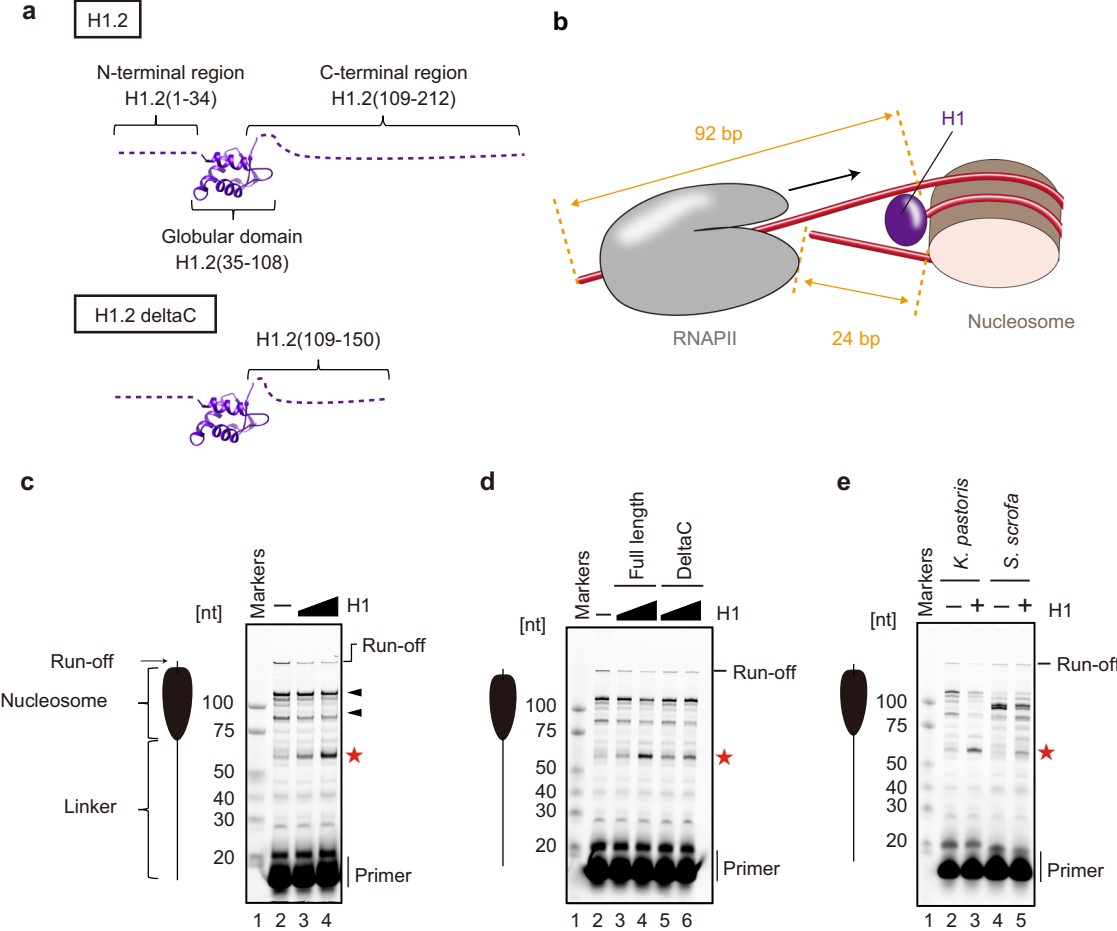

**Fig. 1 | RNAPII transcription on the chromatosome. a** Domain compositions of histone H1.2 and H1.2 deltaC, showing the globular domain and the intrinsically disordered N- and C-terminal regions. **b** Schematic representation of the transcription assay with the chromatosome. **c** Nucleosome transcription assays were conducted in the presence or absence of H1. Elongated RNAs in the nucleosome transcription assay were analyzed by denaturing gel electrophoresis. The nucleosome:H1.2 ratios are 1:0 (lane 2), 1:3 (lane 3), and 1:4.5 (lane 4). Arrowheads indicate major RNAPII pausing points in the nucleosome. The red star indicates the RNA product corresponding to the H1-mediated RNAPII pausing. Reproducibility was confirmed by three independent experiments. **d** Nucleosome transcription assays were conducted in the presence of a C-terminally truncated H1.2 (H1.2 deltaC). Elongated RNAs in the nucleosome transcription assay were analyzed by denaturing gel electrophoresis. The nucleosome:H1.2 ratios are 1:0 (lane 2), 1:3 (lane 3), and 1:4.5 (lane 4). The nucleosome:H1.2 deltaC ratios are 1:3 (lane 5), and 1:4.5 (lane 6). The red star indicates the RNA product corresponding to the H1-mediated RNAPII pausing. Reproducibility was confirmed by six independent experiments. **e** Chromatosome transcription assays were conducted with *S. scrofa* RNAPII (lanes 4 and 5). Elongated RNAs in the nucleosome transcription assay were analyzed by denaturing gel electrophoresis. Lanes 2 and 3 indicate control experiments with *K. pastoris* RNAPII. The nucleosome:H1.2 ratios are 1:0 (lanes 2 and 4) and 1:4.5 (lanes 3 and 5). The red star indicates the RNA product corresponding to the H1-mediated RNAPII pausing. Reproducibility was confirmed by four independent experiments.

Consistent with previous reports of nucleosome transcription with RNAPII[33–35], in the absence of H1, RNAPII transcribed through the nucleosome and paused at the internal nucleosomal DNA positions, especially near the nucleosomal entry and dyad regions (Fig. 1c, lane 2). To avoid undesired H1.2 binding to the bubble DNA region, we added H1.2 after the RNAPII loading (Supplementary Fig. 1b). Interestingly, when H1.2 was added to the reaction, a large amount of a ~60-nt RNA product accumulated. The RNA product increased with increasing amounts of H1.2, indicating that H1.2 causes RNAPII pausing on the entry linker DNA before it enters the nucleosome. The run-off transcription product was substantially reduced, although a certain fraction of RNAPII progressed into the nucleosome in the presence of H1.2 (Fig. 1c, lanes 3 and 4). Therefore, the chromatosome formation by H1.2 recruitment causes RNAPII pausing on the entry linker DNA, and may reduce the overall transcription efficiency through the chromatosome. This is consistent with the previous observation that RNAPII pausing is induced by H1 binding in cellular chromatin[32].

The C-terminal region of H1 reportedly binds to a linker DNA in the chromatosome[9]. We prepared the H1.2 mutant (H1.2 deltaC), lacking half of the C-terminal region (amino acid residues 151-212) (Fig. 1a and Supplementary Fig. 2a). Interestingly, the RNAPII pausing was also observed when the chromatosome transcription was conducted in the presence of H1.2 deltaC (Fig. 1d). The RNAPII pausing by H1.2 deltaC was slightly reduced as compared to that by the full-length H1.2 (Fig. 1d), probably by its defective nucleosome binding. Thus, the H1.2 C-terminal region may not be critical for the H1.2-induced RNAPII pausing.

We next tested whether the RNAPII pausing by the chromatosome formation is conserved in a mammalian RNAPII. To do so, we prepared RNAPII from *Sus scrofa domesticus*[36] (Supplementary Fig. 2b), and performed the chromatosome transcription assay. The reaction conditions were slightly changed from those used with *K. pastoris* RNAPII, because of the lower activity of *S. scrofa* RNAPII. We found that *S. scrofa* RNAPII also paused on the entry linker DNA of the nucleosome in the H1.2-dependent manner (Fig. 1e). This indicated that the H1-induced RNAPII pausing is conserved in a mammalian RNAPII. It should be noted that additional RNAPII pausing inside the nucleosome was observed when the reaction was conducted with the *S. scrofa* RNAPII, although the nucleosomal pausing sites are mostly conserved between the *K. pastoris* and *S. scrofa* RNAPIIs (Fig. 1e, lanes 2 and 4).

### Cryo-EM structures of an RNAPII-chromatosome complex

To reveal the mechanism of the H1-mediated RNAPII pausing in the chromatosome, we performed a cryo-electron microscopy (cryo-EM) analysis (Supplementary Figs. 2e, 3–7 and Supplementary Table 1). The transcription reaction was conducted in the presence of H1.2, and then the RNAPII-chromatosome complexes, in which RNAPII is paused at the entry linker DNA region, were prepared and subjected to data collection by cryo-EM. Three-dimensional (3D) classifications identified two classes of the RNAPII-chromatosome complex structures (forms I and II) paused at the entry linker DNA (Figs. 2 and 3).

The form I complex is the predominant configuration of the RNAPII-chromatosome complex (Fig. 2a). In this complex, the globular domain of H1 binds to the DNA at the nucleosomal dyad (on-dyad binding) (Fig. 2b). The structure of the chromatosome complexed with RNAPII is similar to that of the RNAPII-free chromatosome[8–10]. The H1 binding restricts the orientation and flexibility of the linker DNAs. Surprisingly, we found that the exit linker DNA fixed by H1 collides with the clamp (the Rpb1 clamp core domain) of the RNAPII transcribing the entry linker DNA (Fig. 2c). Interestingly, the exit linker DNA is trapped between the RNAPII clamp and lobe domains (Fig. 2d), and contacts a basic patch (composed of Arg328, Arg329, Arg337, Arg338, Lys340, and Arg341 at the tip of the Rpb2 lobe), as observed in the promoter DNA bound to the preinitiation complex of RNAPII[37] (Supplementary Fig. 8). The H1 globular domain does not directly interact with RNAPII

(Fig. 2b). The specific interaction of RNAPII with the exit linker DNA would hinder the RNAPII progression on the entry linker DNA.

In the form II complex, the exit linker DNA is released from the DNA-binding cleft of RNAPII (Fig. 3a, b), and the chromatosome arrangement relative to the RNAPII is rotated around the entry DNA axis by approximately 65°, as compared to the form I complex (Supplementary Fig. 9). Considering the nucleosome rotation angle, RNAPII may have proceeded ahead by a few base pairs in the form II complex, as compared to the form I complex. Therefore, the form II complex may represent a snapshot structure of the RNAPII just after it escapes from the paused state in front of the chromatosome. Intriguingly, in the form II complex, the RNAPII clamp may directly contact the H1 globular domain because of the RNAPII progression, as compared to the form I complex (Fig. 3c). Further RNAPII progression into the nucleosomal DNA should require the peeling of the entry linker DNA from the H1 in the chromatosome.

### Effect of the downstream nucleosome on the RNAPII pausing

In the RNAPII-chromatosome complex structure (form I), the exit linker DNA of the chromatosome sterically conflicts with the transcribing RNAPII. In the natural chromatin context, this exit linker DNA is followed by another nucleosome (or chromatosome). This led us to test whether a second, neighboring nucleosome/chromatosome affects the RNAPII pausing. To do so, we performed the chromatosome transcription assay with the di-nucleosome template in the presence of H1.2 (Fig. 4a and Supplementary Fig. 10). In this di-nucleosome template, two nucleosomes are connected with a 48 base-pair linker DNA (Fig. 4a), which is an average linker DNA length in transcriptionally active loci in human cells[38]. Similar to the results with the mono-nucleosome template, a ~60-nt RNA product accumulated in the H1.2-dependent manner (Fig. 4b). The structural model suggested that the second nucleosome in the di-nucleosome does not sterically clash with the RNAPII paused on the entry linker DNA of the upstream chromatosome (Supplementary Fig. 11). Therefore, the H1-dependent RNAPII pausing occurs regardless of the presence or absence of the second chromatosome/nucleosome, and the first chromatosome that directly contacts RNAPII plays the major role in impeding transcription elongation.

### Spt4/5 alleviates the H1-mediated RNAPII pausing

Spt4/5 (DSIF for mammals) is one of the transcription elongation factors that is tightly associated with transcribing RNAPII[39,40]. It binds and seals the RNAPII DNA-binding cleft[41], and enhances the transcription elongation efficiency in the nucleosome[35,41–43]. Interestingly, when the Spt4/5-bound RNAPII structure was superimposed on the current RNAPII-chromatosome complex, Spt4/5 overlapped with the exit linker DNA in the form I complex (Fig. 5a). This finding led us to test whether Spt4/5 could alleviate the H1-mediated RNAPII pausing. Our nucleosome transcription assay revealed that the amount of the ~60-nt RNA product from the H1-induced RNAPII pausing at the entry linker DNA is drastically reduced by the addition of Spt4/5 (Fig. 5b, lanes 4 and 9, and Supplementary Fig. 2f). This suggests that Spt4/5 actually alleviates the H1-mediated RNAPII pausing, probably by masking the exit linker DNA binding sites on RNAPII to prevent stable DNA-RNAPII interactions. It should be noted that the ~60-nt RNA product is clearly observed in the presence of Spt4/5, especially at the early reaction time point (5 min) (Fig. 5b, lane 4). Therefore, the H1-mediated RNAPII pausing still occurs in the presence of Spt4/5, although it is drastically alleviated.

## Discussion

This study has revealed that a chromatosome poses a higher transcriptional barrier than an H1-free nucleosome in vitro. H1 causes RNAPII pausing before it enters the chromatosome. This is because H1 brings the entry and exit linker DNAs close together, and the RNAPII on the entry linker DNA sterically conflicts with the exit linker DNA,

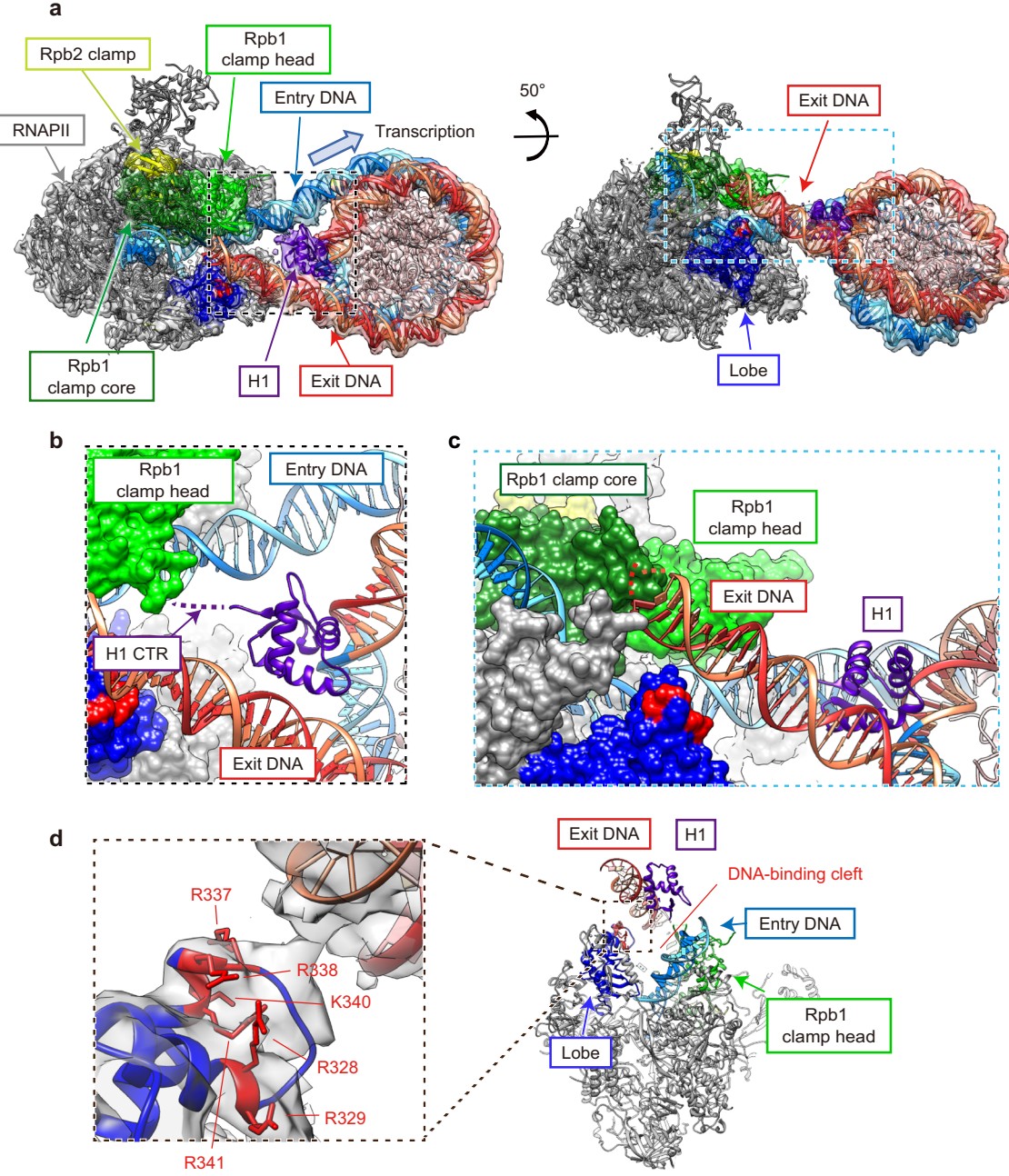

**Fig. 2 | Cryo-EM structure of the RNAPII-chromatosome complex (form I).**
**a** Cryo-EM density of the RNAPII-chromatosome complex (form I) with the fitted structural model. RNAPII, Rpb1 clamp head, Rpb1 clamp core, Rpb2 clamp, lobe, basic patch core histones, and linker histone H1 are colored gray, light green, dark green, yellow, blue, red, pink, and purple, respectively. The entry and exit halves of the nucleosomal DNA relative to the transcribing RNAPII are colored cyan and red, respectively. The regions enclosed by the dashed black and blue squares are presented as close-up views in panels (**b**) and (**c**), respectively. **b** Close-up view of H1 bound to the nucleosome. The predicted H1 C-terminal region (CTR) is indicated by the dashed purple line. **c** Close-up view of the exit linker DNA region clash with RNAPII. The predicted linker DNA regions that are not visualized in the structure are indicated by dotted orange or red lines. **d** The DNA-binding cleft between the RNAPII clamp and lobe domains, and the basic patch (Arg328, Arg329, Arg337, Arg338, Lys340, and Arg341) of the Rpb2 lobe are presented.

hindering its progression. As shown in the form I complex, the exit linker DNA directly interacts with the RNAPII DNA-binding cleft, thus inducing the RNAPII pausing (Fig. 5c, i). The form I complex is the predominant class of RNAPII-chromatosome complexes, suggesting that a majority of RNAPII pauses at this location. It is intriguing that the RNAPII pausing on the entry linker DNA can be weakly detected in the absence of H1.2, although it was drastically enhanced when H1.2 was added (Fig. 1c). One possible explanation is that the exit linker DNA may be captured by the RNAPII DNA-binding cleft even in the absence of H1, although this complex should be unstable because of the

flexibility of the linker DNAs. The H1 binding to the nucleosome would restrict the linker DNA orientations, and may enhance the probability of RNAPII pausing by the exit linker DNA binding to the RNAPII DNA-binding cleft. The transcription elongation factor Spt4/5 may facilitate chromatosome transcription by interfering with the exit linker DNA binding to the RNAPII DNA-binding cleft (Fig. 5a, b).

After the exit linker DNA is released from the RNAPII DNA-binding cleft, RNAPII advances further. RNAPII then pauses again because of the steric clash with the H1 globular domain bound to the nucleosomal dyad, as observed in the form II complex (Fig. 5c, ii). In the form II

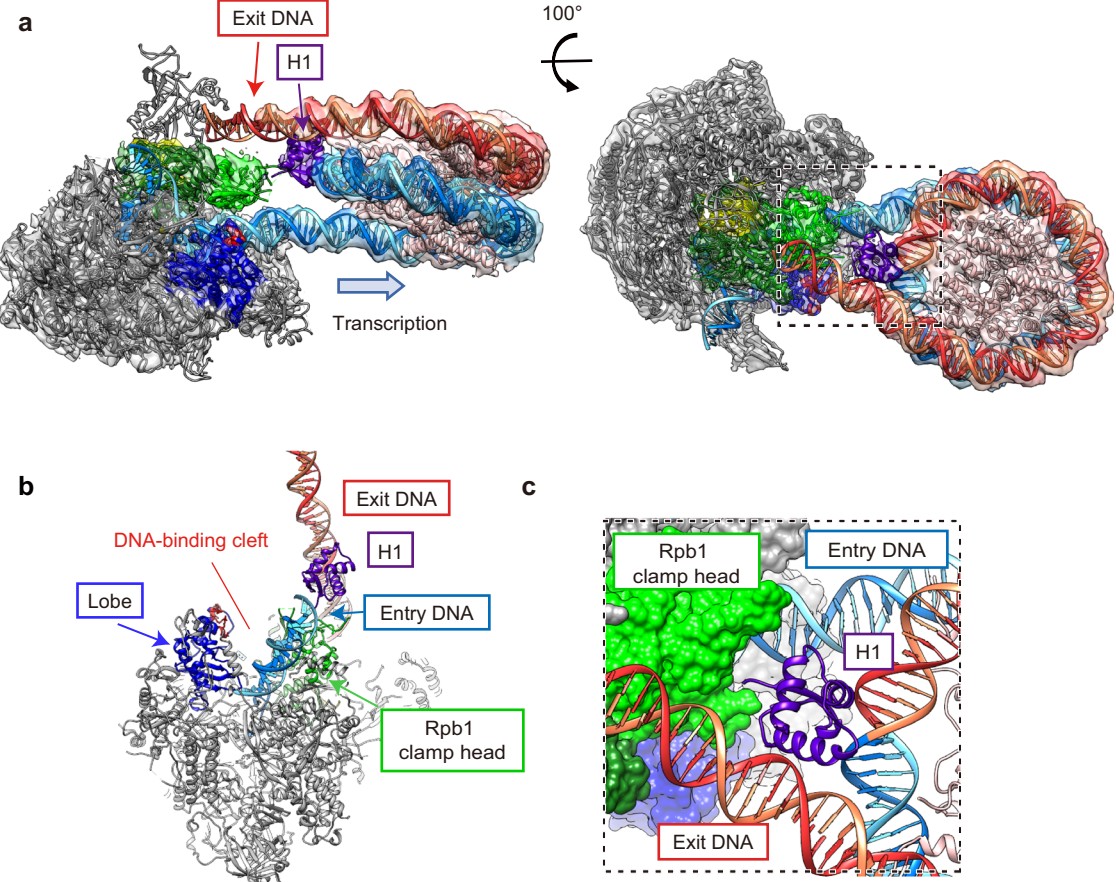

**Fig. 3 | Cryo-EM structure of the RNAPII-chromatosome complex (form II).**
**a** Cryo-EM density of the RNAPII-chromatosome complex (form II) with the fitted structural model. Color codes are the same as those in Fig. 2. The regions enclosed by the dashed black squares are presented as close-up views in panel (**c**). **b** The
DNA-binding cleft between the RNAPII clamp and lobe domains. The exit linker DNA (orange and red) is located outside the DNA-binding cleft of RNAPII. **c** Close-up view of the H1 contact with the Rpb1 clamp head of RNAPII.

complex, the exit linker DNA becomes more flexible as compared to that in the form I complex, by releasing the exit linker DNA from the RNAPII DNA-binding cleft. This may facilitate RNAPII progression in the nucleosomal DNA region. Finally, the steric conflict between RNAPII and the chromatosome may be resolved by peeling the nucleosomal DNA at the entry site (Fig. 5c, iii). A previous study with optical tweezers revealed that the progressive unzipping of the entry linker DNA does not promote H1 dissociation, but repositions it from the on-dyad to off-dyad positions in the chromatosome[14]. Therefore, the bound H1 may persist on the nucleosome while the RNAPII invades the nucleosomal DNA region (Fig. 5c, iii).

In the present study, we have revealed the molecular mechanism by which a single chromatosome impacts transcription by RNAPII. One or more chromatosomes located in certain genomic regions could contribute to regulate the transcription statuses of specific loci, and may function to dictate cell fate during differentiation and tumorigenesis[19–22]. On the other hand, the H1 binding to polynucleosomes causes higher-order chromatin folding, which may serve as a next-level regulatory mechanism for transcription in chromatin[16,44]. Although we examined the effect of Spt4/5 here, transcription elongation is regulated by many additional factors, including transcription elongation factors, histone chaperones, chromatin remodelers, etc.[45,46]. The chromatin structure and the regulatory factors cooperatively regulate the transcription status in cells, and therefore, further structural and biochemical analyses are needed for a better comprehension of the mechanisms.

## Methods
### Preparation of proteins
Human histones H2A, H2B, H3.1, and H4 were expressed in *Escherichia coli* cells, and purified by the method described previously[47,48]. Briefly, the proteins were produced in the BL21(DE3) strain (for H2A, H2B, and H3.1) and the JM109(DE3) strain (for H4) as N-terminally hexa-histidine (His6)-tagged proteins, and isolated under denaturing conditions. The His6-tagged proteins were purified by chromatography with nickel-nitrilotriacetic acid agarose (Ni-NTA) resin (QIAGEN). The His6-tag portion was then removed by thrombin protease, and the histones were purified by cation exchange chromatography on a MonoS column under denaturing conditions. The purified histones were dialyzed against water and lyophilized.

Human H1.2 and H1.2 deltaC (H1.2(1-150)) were expressed in *E. coli* cells and purified by the method described previously[49]. Briefly, H1.2 and H1.2 deltaC were produced in the BL21-CodonPlus(DE3)-RIL strain, as C-terminally His6-SUMO-tagged proteins. The His6-SUMO-tagged proteins were purified by Ni-NTA agarose (QIAGEN) chromatography. The His6-SUMO tag portion was then removed by PreScission protease treatment, and H1.2 and H1.2 deltaC were purified by MonoS cation exchange column chromatography. Purified H1.2 and H1.2 deltaC were stored at −80 °C. *K. pastoris* RNAPII, composed of 12 subunits, was purified as described previously[40,50]. Briefly, *K. pastoris* encoding a TAP-tagged Rpb2 was cultured, and the tagged-RNAPII was purified by affinity column chromatography on anti-FLAG M2 affinity gel (Sigma-Aldrich), and anion-exchange column chromatography. *K. pastoris*

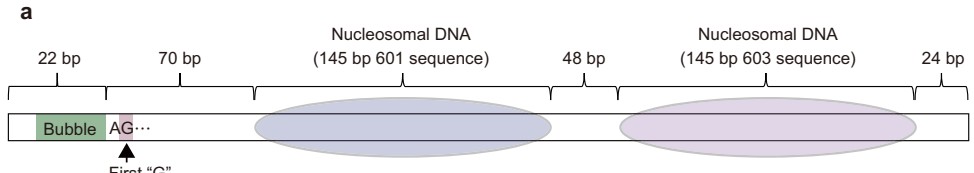

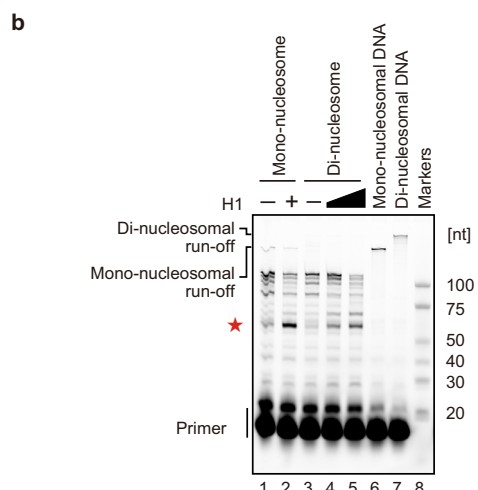

**Fig. 4 | RNAPII transcription on the di-chromatosome. a** Graphical representation of the template for di-chromatosome transcription by RNAPII. **b** The transcription assays were conducted in the presence or absence of H1. Elongated RNAs were analyzed by denaturing gel electrophoresis. The mono-nucleosome:H1.2

ratios are 1:0 (lane 1), and 1:4.5 (lane 2). The di-nucleosome:H1.2 ratios are 1:0 (lane 3), 1:6 (lane 4), and 1:9 (lane 5). The red star indicates the RNA product corresponding to the H1-mediated RNAPII pausing. Reproducibility was confirmed by four independent experiments.

TFIIS and Spt4/5 were purified as described previously[40]. These proteins were produced as His6-tagged proteins in the *E. coli* KRX strain. The His6-tagged proteins were then purified by chromatography on a COSMOGEL His-Accept column (Nacalai Tesque), and eluted by cleavage of the His6-tag portion by HRV-3C protease treatment. The eluted proteins were purified by Resource S cation-exchange column chromatography.

For the *S. scrofa* RNAPII preparation, fresh *S. scrofa* thymus was homogenized using a 2L blender (Waring) in buffer A (50 mM Tris-HCl (pH 7.9), 1 mM EDTA, 10 μM ZnCl$_2$, 10% glycerol, 1 mM DTT, 1 mM phenylmethylsulfonyl fluoride, 1 mM benzamidine, 0.6 μM leupeptin, 2 μM pepstatin A). The homogenized material was centrifuged at 12,200 × g for 20 min at 4 °C, and the supernatant was filtered through Miracloth (Millipore). A 5% solution of polyethyleneimine (adjusted to pH 7.9 at 25 °C) was added to a final concentration of 0.1%. After mixing for 10 min at 4 °C, the precipitate was collected by centrifugation at 15,750 xg for 20 min. The pellet was washed twice with buffer A, and the RNAP II fraction was extracted from the pellet with buffer A containing 0.15–0.2 M ammonium sulfate. A Q Sepharose (Cytiva) slurry was added to the eluted fraction, and the ammonium sulfate concentration was adjusted to 0.15 M. After rotation at 4 °C for 30 min, the resin was washed with 6 volumes of buffer A containing 0.15 M ammonium sulfate, followed by 3 volumes of buffer A containing 0.2 M ammonium sulfate. RNAPII was eluted with buffer A containing 0.4 M ammonium sulfate. The eluate was precipitated by 50% ammonium sulfate overnight. The precipitate was collected by centrifugation at 22,900 × g for 20 min at 4 °C, resuspended in buffer A, and then dialyzed against 1 L of buffer B (50 mM Tris-HCl (pH 7.9), 150 mM NaCl, 1 mM EDTA, 10 μM ZnCl$_2$, 2 mM DTT). The dialyzed sample was subjected to anion-exchange chromatography on a Mono Q (Cytiva) column. The RNAPII-containing fraction was concentrated with an Amicon Ultra-15 centrifugal filter unit (Merck) and purified by size

exclusion chromatography on a Superose 6 column (Cytiva), equilibrated with buffer C (5 mM HEPES-NaOH (pH 7.3), 150 mM NaCl, 10 μM ZnCl$_2$, 10 mM DTT). Peak fractions containing RNAPII were collected and stored as a 10% glycerol stock at −80 °C. Judging from the electrophoretic band intensities, the purity of RNAPII in this fraction is about 30% (Supplementary Fig. 2b).

For the *H. sapiens* TFIIS preparation, the DNA fragment encoding TFIIS was inserted into the pET15b vector. TFIIS was expressed in *E. coli* as an N-terminally hexa-histidine tagged protein, and purified by Ni-NTA affinity column chromatography. After tag cleavage by HRV3C protease, the resulting TFIIS protein was further purified by MonoS cation exchange column chromatography. The purified TFIIS protein was dialyzed against 20 mM HEPES-NaOH (pH 7.5) buffer, containing 150 mM NaCl, 1 mM DTT, 5% glycerol, and 10 μM zinc chloride, flash-frozen in liquid nitrogen, and stored at −80 °C.

**Preparation of the DNA fragment for chromatosome assembly**
For the preparation of the DNA fragment used in the mono-chromatosome transcription, the plasmid DNA containing the 193 bp Widom 601 sequence was prepared from the *E. coli* DH5 strain. The 193 bp Widom 601 fragment was excised from the plasmid DNA by EcoRV (Takara), and purified by polyethylene glycol precipitation. The ends of the DNA fragments were dephosphorylated by calf intestinal alkaline phosphatase (Takara). The 193 bp Widom 601 DNA fragment was also produced by polymerase chain reaction. The purified 193 bp Widom 601 DNA fragment was cleaved with HinfI, resulting in the 161 base-pair DNA with a 3 bp overhang at the 5′ end (Takara). This DNA fragment was further purified by DEAE-5PW anion-exchange column chromatography (TOSOH) or native polyacrylamide gel electrophoresis purification using a Prep Cell model 491 apparatus (Bio-Rad).

The sequences of both strands of the resulting DNA fragment are as follows:

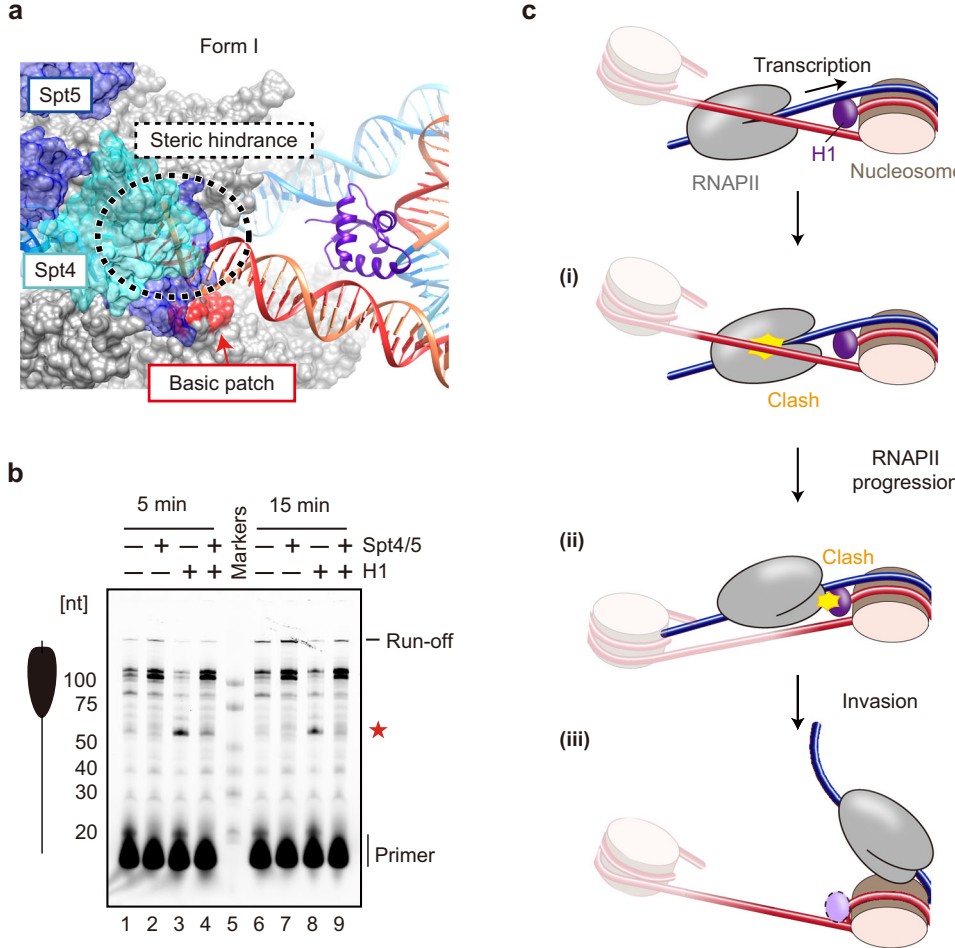

**Fig. 5 | Effect of Spt4/5 on chromatosome transcription by RNAPII.**
**a** Superimposition of Spt4/5 in the Spt4/5-RNAPII complex (PDB ID: 5OIK) on the form I complex. Color codes are the same as those in Fig. 2, except Spt4 and Spt5 are colored cyan and dark blue, respectively. **b** The nucleosome transcription assay was conducted in the presence or absence of H1 and Spt4/5. Elongated RNAs in the nucleosome transcription assay were analyzed by denaturing gel electrophoresis. The nucleosome:H1.2 ratios are 1:0 (lanes 1, 2, 6, and 7), and 1:4.5 (lanes 3, 4, 8, and 9). The nucleosome:Spt4/5 ratios are 1:0 (lanes 1, 3, 6, and 8), and 1:4 (lanes 2, 4, 7, and 9). The red star indicates the RNA product corresponding to the H1-mediated RNAPII pausing. Reproducibility was confirmed by three independent experiments. **c** A model for RNAPII progression on the chromatosome. The RNAPII transcribes

the entry linker DNA in front of the chromatosome (Top), and pauses when the exit linker DNA is captured by the RNAPII DNA-binding cleft, as revealed by the form I complex (i). The exit linker DNA is released from the RNAPII DNA-binding cleft by the RNAPII progression, and the RNAPII is paused again by the steric clash with the H1 globular domain bound to the nucleosomal dyad, as revealed by the form II complex (ii). The exit linker DNA of the form II complex is free from the RNAPII DNA-binding cleft, and is more flexible than that of the form I complex. The entry DNA region may be peeled from H1 and the histone surface of the chromatosome by the RNAPII progression, and the steric conflict between RNAPII and the chromatosome may be resolved (iii).

non-template strand: 5′:
AATCCGGTGCCGAGGCCGCTCAATTGGTCGTAGACAGCTCTAG
CACCGCTTAAACGCACGTACGCGCTGTCCCCCGCGTTTTAACCGCCA
AGGGGATTACTCCCTAGTCTCCAGGCACGTGTCAGATATATACATCC
AGGCCTTGTGTCGCGAAATTCATAGAT

template strand: 5′:
ATCTATGAATTTCGCGACACAAGGCCTGGATGTATATATCTGAC
ACGTGCCTGGA-
GACTAGGGAGTAATCCCCTTGGCGGTTAAAACGCGGGGGACAGCG
CGTACGTGCGTTTAAGCGGTGCTAGAGCTGTCTACGACCAATTGAGC
GGCCTCGGCACCGG.

The purified DNA fragment was ligated with the double-stranded oligonucleotide 50-mer (purchased from Eurofins) by T4 DNA ligase (NIPPON GENE). The sequences of both strands of the double-stranded oligonucleotide 50-mer are as follows:
non-template strand: 5′:
CCTTTAAAGCAATAGGAGCTTACGGTCCACTTGTGTTTGGTGTG
TTTGGG
template strand: 5′:

ATTCCCAAACACACCAAACACAAGTGGACCGTAAGCTCCTATT
GCTTTAA.

For the preparation of the DNA fragment used in the di-chromatosome transcription, the 368 bp DNA fragment containing the Widom 601 and 603 sequences connected with 48 bp linker DNA was produced by polymerase chain reaction. The purified DNA fragment was cleaved with HinfI, resulting in the 354 bp DNA with a 3 bp overhang at the 5′ end (Takara). This DNA fragment was further purified by native polyacrylamide gel electrophoresis, using a Prep Cell model 491 apparatus (Bio-Rad).

The sequences of both strands of the resulting DNA fragment are as follows:
non-template strand: 5′:
AATCCGGTGCCGAGGCCGCTCAATTGGTCGTAGACAGCTCTAG
CACCGCTTAAACGCACGTACGCGCTGTCCCCCGCGTTTTAACCGCCA
AGGGGATTACTCCCTAGTCTCCAGGCACGTGTCAGATATATACATCC
AGGTCATTCCGGACGTGTTTGTCCTCTGCCTTTAAAGCAATAGGAGC
TTACCCCCAGGGACTTGAAGTAATAAGGACGGAGGGCCTCTTTCAAC
ATCGATGCACGGTGGTTAGCCTTGGATTGCGCTCTACCGTGCGCTAA

GCGTACTTAGAAGCCCGAGTGACGACTTCACACGGTAGGTGGGC
GCGCGAACTGAGCCTTGTGTCGCGAAATTCATGAT

template strand: 5′:
ATCATGAATTTCGCGACACAAGGCTCAGTTCGCGCGCCCAC
CTACCGTGTGAAGTCGTCACTCGGGCTTCTAAGTACGCTTAGCGCA
CGGTAGAGCGCAATCCAAGGCTAACCACCGTGCATCGATGTTGAAAG
AGGCCCTCCGTCCTTATTACTTCAAGTCCCTGGGGGTAAGCTCCTAT
TGCTTTAAAGGCAGAGGACAAACACGTCCGGAATGACCTGGATGTAT
ATATCTGACACGTGCCTGGAGACTAGGGAGTAATCCCCTTGGCGGT
TAAAACGCGGGGGACAGCGCGTACGTGCGTTTAAGCGGTGCTAGAGC
TGTCTACGACCAATTGAGCGGCCTCGGCACCGG.

The purified DNA fragment was ligated with the same double-stranded oligonucleotide 50-mer (purchased from Eurofins) as described above by T4 DNA ligase (NIPPON GENE).

## Reconstitution of the nucleosome

The histone octamer was reconstituted as described previously[47,48]. Briefly, the freeze-dried histones were mixed at an equal molar ratio under denaturing conditions. The histone octamer was refolded by dialysis against refolding buffer, containing 10 mM Tris-HCl (pH 7.5), 1 mM EDTA, 2 M NaCl, and 5 mM 2-mercaptoethanol. The histone octamer was purified by chromatography on a Superdex 200 gel filtration column (GE Healthcare). The resulting octamer was flash-frozen in liquid nitrogen, and stored at −80 °C. The histone octamer and the 161 or 354 bp DNA fragment were mixed, and the nucleosome was reconstituted by the salt dialysis method. The double-stranded oligonucleotide 47-mer containing a 3 bp overhang at the 5′ end was ligated to the cohesive DNA end of the reconstituted nucleosome. The oligonucleotide also contained a 9 base mismatched region (bubble) at the region 14–22 bps from the 3′ end. The resulting nucleosome with a bubble region was purified by non-denaturing gel electrophoresis, using a Prep Cell apparatus (Bio-Rad).

The sequences of both strands of the oligonucleotide 47-mer containing a bubble region are as follows:

non-template strand: 5′
TTCTTAAATACCATGGCCATCTTCATTCCGGACGTGTTTGTC
CTCTG

template strand: 5′:
AGGCAGAGGACAAACACGTCCGGAATGAGAGCTAATTTGGTATT
TAAGAA.

The complete DNA sequences of both strands for the mono-chromatosome transcription experiments are as follows:

non-template strand: 5′
TTCTTAAATACCATGGCCATCTTCATTCCGGACGTGTTTGTCCTC
TGCCTTTAAAGCAATAGGAGCTTACGGTCCACTTGTGTTTGGTGTG
TTTGGGAATCCGGTGCCGAGGCCGCTCAATTGGTCGTAGACAGCTCT
AGCACCGCTTAAACGCACGTACGCGCTGTCCCCCGCGTTTTAACCG
CCAAGGGGATTACTCCCTAGTCTCCAGGCACGTGTCAGATATATACA
TCCAGGCCTTGTGTCGCGAAATTCATAGAT

template strand: 5′
ATCTATGAATTTCGCGACACAAGGCCTGGATGTATATATCTGAC
ACGTGCCTGGAGACTAGGGAGTAATCCCCTTGGCGGTTAAAACGCGG
GGGACAGCGCGTACGTGCGTTTAAGCGGTGCTAGAGCTGTCTACG
ACCAATTGAGCGGCCTCGGCACCGGATTCCCAAACACACCCAAACACA
AGTGGACCGTAAGCTCCTATTGCTTTAAAGGCAGAGGACAAACACGT
CCGGAATGAGAGCTAATTTGGTATTTAAGAA.

The complete DNA sequences of both strands for the di-chromatosome transcription experiments are as follows:

non-template strand: 5′
TTCTTAAATACCATGGCCATCTTCATTCCGGACGTGTTTGT
CCTCTGCCTTTAAAGCAATAGGAGCTTACGGTCCACTTGTGTTTGG
TGTGTTTGGGAATCCGGTGCCGAGGCCGCTCAATTGGTCGTAGACAG
CTCTAGCACCGCTTAAACGCACGTACGCGCTGTCCCCCGCGTTTTAA
CCGCCAAGGGGATTACTCCCTAGTCTCCAGGCACGTGTCAGATATAT
ACATCCAGGTCATTCCGGACGTGTTTGTCCTCTGCCTTTAAAGCAAT

AGGAGCTTACCCCCAGGGACTTGAAGTAATAAGGACGGAGGGCCT
CTTTCAACATCGATGCACGGTGGTTAGCCTTGGATTGCGCTCTA
CCGTGCGCTAAGCGTACTTAGAAGCCCGAGTGACGACTTCACACG
GTAGGTGGGCGCGCGAACTGAGCCTTGTGTCGCGAAATTCATGAT

template strand: 5′
ATCATGAATTTCGCGACACAAGGCTCAGTTCGCGCGCCCACCTA
CCGTGTGAAGTCGTCACTCGGGCTTCTAAGTACGCTTAGCGCACGGT
AGAGCGCAATCCAAGGCTAACCACCGTGCATCGATGTTGAAAGAGGC
CCTCCGTCCTTATTACTTCAAGTCCCTGGGGGTAAGCTCCTATTGCT
TTAAAGGCAGAGGACAAACACGTCCGGAATGACCTGGATGTATATAT
CTGACACGTGCCTGGAGACTAGGGAGTAATCCCCTTGGCGGTTAAAA
CGCGGGGGACAGCGCGTACGTGCGTTTAAGCGGTGCTAGAGCTGTCT
ACGACCAATTGAGCGGCCTCGGCACCGGATTCCCAAACACACCAAAC
ACAAGTGGACCGTAAGCTCCTATTGCTTTAAAGGCAGAGGACAAACA
CGTCCGGAATGAGAGCTAATTTGGTATTTAAGAA

## Transcription assay on the chromatosome

The mono-nucleosome or di-nucleosome (0.14 µM) was incubated with RNAPII (0.14 µM of *K. pastoris* RNAPII or approximately the corresponding amount of the partially purified *S. scrofa* RNAPII), TFIIS (0.14 µM of *K. pastoris* TFIIS or 0.11 µM of *H. sapiens* TFIIS), and the DY647-labeled primer RNA (5′-DY647- AUAAUUAGCUC-3′) (0.57 µM) (Dharmacon), in the presence or absence of Spt4/5 (0.57 µM), in a 7 µL reaction at 30 °C for 30 min. For the experiment with *K. pastoris* RNAPII, the reaction solution contained 37 mM HEPES-KOH (pH 7.5), 7.1 mM MgCl₂, 43 mM potassium acetate, 0.29 µM zinc acetate (0.018 µM for the experiments in Fig. 1c), 29 µM Tris(2-carboxyethyl) phosphine, 0.14 mM DTT, 2.1% glycerol, 0.57 mM UTP, 0.57 mM GTP, and 0.57 mM ATP. For *S. scrofa* RNAPII, the reaction solution contained 36 mM HEPES-KOH (pH 7.5), 0.37 mM HEPES-NaOH (pH 7.5), 7.1 mM MgCl₂, 37 mM potassium acetate, 0.29 µM zinc acetate, 29 µM Tris(2-carboxyethyl)phosphine, 0.46 mM DTT, 2.0% glycerol, 6.14 mM NaCl, 0.29 µM zinc chloride, 0.57 mM UTP, 0.57 mM GTP, and 0.57 mM ATP. The reactions were then incubated for 15 min at 4 °C for transcription with *K. pastoris* RNAPII or 30 °C for transcription with *S. scrofa* RNAPII. H1.2 was added to the reaction mixture in nucleosome:H1.2 ratios of 1:0, 1:3, 1:4.5, 1:6, and 1:9, and was bound to the nucleosome for 30 min by an incubation at 4 °C for transcription with *K. pastoris* RNAPII or at 30 °C for transcription with *S. scrofa* RNAPII. H1.2 was dissolved in 10 mM Tris-HCl buffer (pH 7.5), 0.05 mM phenylmethylsulfonyl fluoride (PMSF), 0.25 mM EDTA, 75 mM NaCl, 10% glycerol, 10 mM HEPES-KOH (pH 7.5), 0.5 mM dithiothreitol, 50 mM KCl, and 1 mM 2-mercaptoethanol. CTP (1 µL, final 0.4 mM) was then added to re-start the RNAPII progression. After the reaction (Figs. 1c, d and 5b: 15 min; Figs. 1e, 4b, and 5b: 5 min), each aliquot (2 µL) was stopped by adding 1 µL of stop solution (200 mM Tris-HCl (pH 8.0), 0.5 mg/mL Proteinase K (Roche), 0.25% SDS, and 80 mM EDTA). The samples were kept at room temperature for 10 min. Hi-Di formamide (12 µL, Thermo Fisher Scientific) was added to the samples, which were then heated at 95 °C. The resulting elongated RNAs were analyzed by 10% denaturing polyacrylamide gel electrophoresis. Uncropped gels are shown in Supplementary Fig. 12.

## Preparation of the RNAPII-chromatosome complexes for cryo-EM

The 6-FAM-labeled nucleosome (0.14 µM) was mixed with RNAPII (0.14 µM), TFIIS (0.14 µM), and the DY647-labeled primer RNA (5′-DY647- AUAAUUAGCUC-3′) (0.57 µM) (Dharmacon) in 1.5 mL of reaction solution, containing 37 mM HEPES-KOH (pH 7.5), 7.1 mM MgCl₂, 4.3 mM potassium acetate, 0.29 µM zinc acetate, 29 µM Tris(2-carboxyethyl)phosphine, 0.14 mM DTT, 0.7% glycerol, 0.57 mM UTP, 0.57 mM GTP, and 0.57 mM ATP, at 30 °C for 30 min, followed by an incubation at 4 °C for 15 min. Afterwards, H1.2 was added to the reaction mixture at a nucleosome:H1.2 ratio of 1:6, and was bound to the nucleosome by an incubation at 4 °C for 30 min. CTP (216 µL, final

0.4 mM) was then added to re-start the RNAPII progression. After a 5 min reaction, 216 μL of 0.5 M EDTA was added to the reaction mixture. Under these conditions, a ~60-nt RNA product predominantly accumulated in the presence of H1.2. The resulting RNAPII-chromatosome complexes were fractionated by the GraFix method[51]. A gradient was prepared with low buffer (20 mM HEPES·KOH (pH 7.5), 50 mM potassium acetate, 0.2 μM zinc acetate, 0.1 mM Tris(2-carboxyethyl)phosphine, and 10% (w/v) sucrose) and high buffer (20 mM HEPES·KOH (pH 7.5), 50 mM potassium acetate, 0.2 μM zinc acetate, 0.1 mM Tris(2-carboxyethyl)phosphine, 25% (w/v) sucrose, and 0.1% glutaraldehyde). The sample was applied to the top of the gradient and centrifuged at 4 °C for 16 h at 124,779 × $g$, using the SW41 rotor (Beckman Coulter). After centrifugation, the fractions containing the RNAPII-chromatosome complexes were collected and dialyzed twice against 20 mM HEPES-NaOH (pH 7.5) buffer, containing 0.2 μM zinc acetate and 0.1 mM Tris(2-carboxyethyl)phosphine. The resulting RNAPII-chromatosome complexes were then concentrated with an Amicon Ultra 100 K centrifugal filter unit (Millipore), until the DNA concentration of the sample reached 78.4 μg/mL. The samples were applied to glow-discharged Quantifoil grids (R1.2/1.3, Cu, 200 mesh; Quantifoil). Each grid was blotted at 4 °C and 100% humidity in a Vitrobot Mark IV (Thermo Fisher Scientific) (blot force 5; blot time 3 s), and then immediately plunged into liquid ethane.

### Cryo-EM data collection
Cryo-EM images of the RNAPII-chromatosome complexes were collected by a Krios G3i electron microscope (Thermo Fisher Scientific), equipped with a K3 direct electron detector with a 25 eV slit width of the BioQuantum energy filter (Gatan), and operated at 300 kV at a pixel size of 1.07 Å. Each image of the RNAPII-chromatosome complex was recorded with a 7 s exposure time, and then fractionated into 40 frames with a total dose of 56.2 e⁻/Å, using the SerialEM software[52].

### Cryo-EM image processing
First, the motion correction and CTF estimation were performed with Relion3[53] and Gctf[54], respectively. Preliminary RNAPII-chromatosome maps were then obtained after particle picking with gautomatch (http://www.mrc-lmb.cam.ac.uk/kzhang/) and quick 2D and 3D classifications with Relion3. Using the particle coordinates from these preliminary processing results, the neural networks for Topaz[55] were trained, and then another round of particle picking was performed with Topaz. Finally, the coordinates from the Topaz picking and the initial gautomatch picking were merged, and served as the starting set for further image processing with Relion3.

During the initial stage of image processing, the dataset was split in two batches that were processed independently. As the dataset was highly heterogeneous, refinements were mostly focused on the RNAPII region of the molecule, and to remove bad particles and particles centered on the nucleosome, 2D and 3D classifications were performed occasionally. After the initial 3D refinement using the RNAPII mask, the Bayesian polishing and CTF refinement were performed. After another 3D refinement, 3D classification was performed to remove RNAPII particles lacking downstream DNA. A 2.8 Å resolution RNAPII map was obtained at this stage, and then another Bayesian polishing was performed to downscale particles to 1.49 Å/pix. Finally, 3D classifications with the mask enveloping RNAPII and nucleosome were done to select particles containing both RNAPII and nucleosome.

Next, the particles from the two batches were merged and subjected to 3D classification, which revealed two different complexes of RNAPII transcribing a chromatosome (forms I and II). For each complex, RNAPII subtraction and 3D classification around the nucleosome were performed to select classes with high quality chromatosome density. A significant amount of continuous motion remained between the RNAPII and chromatosome, so local refinements for the RNAPII region and the chromatosome region were performed, in addition to

the overall reconstruction. Composite maps were also calculated for each complex structure, using the phenix.combine_focused_maps tool in the Phenix package[56].

### Model building
First, the RNAPII and the chromatosome were independently modeled. The atomic model for RNAPII was based on the previous crystal structure (PDB: 5XOG)[40], modeled manually with WinCoot (COOT[57] for windows), and then refined against the consensus RNAPII reconstruction using phenix.real_space_refine tool in the Phenix package[56]. The atomic model for the chromatosome was based on previous crystal structures (PDB: 3LZ0 and 4QLC)[8,58], modeled manually, and then refined against the nucleosome reconstruction of the form I complex. The RNAPII and nucleosome models were then fit into their respective overall reconstructions, and the interfaces between the two bodies and the connecting DNA were manually edited with WinCoot and ISOLDE[59]. Figures were prepared using UCSF Chimera[60], ChimeraX[61], and PyMOL.

### Reporting summary
Further information on research design is available in the Nature Portfolio Reporting Summary linked to this article.

## Data availability
The data that support this study are available from the corresponding authors upon reasonable request. The cryo-EM reconstructions and atomic models of the RNA polymerase II-nucleosome-H1 complexes generated in this study have been deposited in the Electron Microscopy Data Bank and the Protein Data Bank, under the accession codes EMD-34415 and PDB ID 8H0V for the RNAPII-nucleosome-H1 form I complex, and EMD-34416 and PDB ID 8H0W for the RNAPII-nucleosome H1 form II complex. The structures used in this study can be found in the Protein Data Bank under the accession codes: 5OIK, 5XOG, 3LZ0, 4QLC, and 7K5Y.

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

## Acknowledgements

We thank M. Ogasawara (Univ. of Tokyo) and Dr. M. Kikkawa (Univ. Tokyo) for cryo-EM data collection, and Y. Iikura, M. Dacher, and Y. Takeda (Univ. of Tokyo) for their assistance. We thank J. Kato (Univ. of Tokyo), M. Naganuma, M. Goto, M. Aoki, and M. Henmi (RIKEN) for protein preparation. This work was supported in part by JSPS KAKENHI Grant Numbers JP20H03201 [to H.E. and T.K.], JP18H05534 [to H.K.], JP20H00449 [to H.K.], JP20H05690 [to T.K. and S.S.], and JP22K06098 [to Y.T.], Research Support Project for Life Science and Drug Discovery (BINDS) from AMED under Grant Numbers JP22ama121009 [to H.K.] and JP22ama121002j001 [to M. Kikkawa, Univ. of Tokyo], and support by JST ERATO Grant Number JPMJER1901 [to H.K.].

## Author contributions

R.H. and T.K. prepared the RNAP II-nucleosome-H1 complexes and performed biochemical analyses. H.E., T.K., Y.T., and S.S. performed cryo-EM analyses. T.U. prepared *S. scrofa* RNAPII. S.S. and H.K. conceived, designed, and supervised all of the work. R.H., H.E., Y.T., S.S., and H.K. prepared all figures and wrote the paper. All of the authors discussed the results and commented on the manuscript.

## Competing interests

The authors declare no competing interests.
