## [Peer Review File · Nature Communications]

REVIEWER COMMENTS

Reviewer #1 (Remarks to the Author):

In this manuscript, Rina Hirano et al. determined the Cryo-EM structures of transcribing Pol II-chromatosome complexes in two forms. The two structures provide the mechanism of H1 pauses Pol II transcript and show the sequential process of Pol II transcript on the chromatosome. While the results are interesting, the manuscript was not well prepared and should be largely improved before publication.

Major concerns:

1. The H1 is commonly thought to compact chromatin and repress transcription. Although the H1 function is critical for transcriptional regulation, a direct connection between H1-containing nucleosome and Pol II elongation complex remains controversial. The authors should discuss previous finding and clarify this issue in their manuscript to avoid misleading to readers.
2. The authors should fit their structures and working model into nucleosome array, which represent the natural chromatin with evenly-spaced nucleosomes (linker DNA ~20 to 30 bp). For example, there is no exposed end in cells when Pol II elongation complex proceeds on chromatin (with H1 or not). In the complex structure in form-1, the exit DNA generates clash with Pol II. From the figure, I can't see the clash in detail. The authors may describe what the structure/complex would be if the exit DNA has no exposed end and if the exit end associates with a nucleosome. This illustration will clarify whether this conformation fits the complex in the physiological context.
3. Fig. 1c shows that there are nearly half of Pol II transcribed into the nucleosome in the presence of H1. The result seems to be inconsistent with the structural observation and the conclusion of the manuscript. Did the author observe particles showing that Pol II enters into the nucleosome?
4. The figures, main text (including the discussion) were not well prepared and should be improved before publication.

Minor concerns:

1. Fig. 2a, Fig.3a, and 3b are not very clear. The authors should consider changing the model color, or the model display.
2. The assembly of RNAPII-chromatosome complexes. Why not assembled the chromatosome before loading the pol II on the template nucleosome?
3. The authors should show the SDS-PAGE of purified proteins, and angular distributions and local resolution estimation of the cryo-EM reconstructions.
4. Why did the authors choose histone H1.2? As far as I know, H1.2, which has only weak chromatin compaction ability and H1.2 does not involve direct modulation of chromatin structure.

Reviewer #2 (Remarks to the Author):

In this manuscript, the authors reported a structural study on how linker histone H1 inhibits the transcription function of RNA polymerase II (RNAPII). The system consists of a nucleosome bound to H1 with two flanking/linker DNA (92 bp and 24 bp). The longer

linker DNA can form a transcription bubble with a primer RNA and is considered the entry DNA, while the other is the exit DNA. Transcription reaction shows that H1 help accumulate a ~60-nt RNA product. They then visualized the structures of RNAPII bound to the nucleosome with H1 and identified two structural forms (Form I and Form II) that have RNAPII paused at the entry DNA before H1. RNAPII moves ~2bp further on the entry DNA than Form I.

Linker histones have been proposed to inhibit transcription. However, no structure is available to help illustrate how it happens. The two structures presented in this manuscript provide the first structural snapshots of how H1 may inhibit transcription. This study should be of great interest to researchers in the fields of transcription and chromatin.

Some issues need to be clarified.

1. The authors used yeast *Komagataella pastoris* (belongs to the family of budding yeast) RNAPII and human linker histone. Budding yeast chromatin has much shorter linker DNA and no canonical H1. I suppose the authors are trying to mimic high-order systems in this study. If so, they should test if a similar reaction product is produced using mammalian RNAPII. At least the authors should discuss the rationale for choosing this hybrid system, for example, by comparing the structures of mammalian and yeast RNAPII.
2. Two bands at ~60-nt exist in the absence of H1, suggesting that the product is not determined by the specific interactions between RNAPII and H1. It needs to be pointed out.
3. The structure form I shows that the linker DNA blocks the RNAPII. It is hypothesized this is the mechanism for the inhibition. The two DNA linkers is held together by linker histone tails. It would be important to verify the hypothesis by studying the effects of removing the H1 C-terminal domain on transcription.
4. Related to 3, this study would benefit from the inclusion of RNAPII elongation studies performed with di-nucleosomes.
5. If the authors believe the basic patch is critical for pausing, mutations that remove the basic patch should be made to test their hypothesis (if the experiment is doable.)

REVIEWER COMMENTS

Reviewer #1 (Remarks to the Author):

In this manuscript, Rina Hirano et al. determined the Cryo-EM structures of transcribing Pol II-chromatosome complexes in two forms. The two structures provide the mechanism of H1 pauses Pol II transcript and show the sequential process of Pol II transcript on the chromatosome. While the results are interesting, the manuscript was not well prepared and should be largely improved before publication.

Reply)

Thank you very much for your favorable evaluation. We have revised the manuscript according to your suggestions.

Major concerns:

1. The H1 is commonly thought to compact chromatin and repress transcription. Although the H1 function is critical for transcriptional regulation, a direct connection between H1-containing nucleosome and Pol II elongation complex remains controversial. The authors should discuss previous finding and clarify this issue in their manuscript to avoid misleading to readers.

Reply)

Thank you very much for this comment. In the revised manuscript, we explained that most H1 subtypes function as transcription suppressors in the Introduction, and added previously obtained information about H1.2 as a transcription suppressor and activator with new citations (p.3, ll.55-57 and p.4, ll.66-76). These additional explanations will avoid misleading the readers.

2. The authors should fit their structures and working model into nucleosome array, which represent the natural chromatin with evenly-spaced nucleosomes (linker DNA ~20 to 30 bp). For example, there is no exposed end in cells when Pol II elongation complex proceeds on chromatin (with H1 or not). In the complex structure in form-1, the exit DNA generates clash with Pol II. From the figure, I can't see the clash in detail. The authors

may describe what the structure/complex would be if the exit DNA has no exposed end and if the exit end associates with a nucleosome. This illustration will clarify whether this conformation fits the complex in the physiological context.

Reply)

Thanks again for this comment. In the revised manuscript, we illustrated an RNAPII transcription model with a di-nucleosome, in which two nucleosomes are connected with an appropriate length (48 base pairs) of a linker DNA (corresponding to the exit linker DNA for our current RNAPII-chromatosome structures) in the new Extended Data Fig. 11. The 48 base-pair linker DNA is known as an average linker DNA length in transcriptionally active loci in human cells. In addition, we performed the chromatosome transcription assay with the di-nucleosome, and found that the RNAPII pausing is still induced on the entry linker DNA of the upstream nucleosome by H1 binding. These new results are presented in the new Fig. 4, and described in the new Results section “The downstream nucleosome does not affect the H1-dependent RNAPII pausing” (p.8, l.167-p.9, l.183).

3. Fig. 1c shows that there are nearly half of Pol II transcribed into the nucleosome in the presence of H1. The result seems to be inconsistent with the structural observation and the conclusion of the manuscript. Did the author observe particles showing that Pol II enters into the nucleosome?

Reply)

As this reviewer pointed out, the RNAPII transcribed into the nucleosome in the presence of H1. We described this fact in the text (p. 5, ll.104-106). In addition, we have tried to capture the RNAPII-chromatosome particles, in which RNAPII enters into the nucleosome. However, we could not detect such particles in our cryo-EM images. This should be an interesting project in the future.

4. The figures, main text (including the discussion) were not well prepared and should be improved before publication.

Reply)

In the revised manuscript, we extensively revised the figures and text according to this reviewer's suggestions. Please find these changes in the revised text and figures. These changes are explained in our responses to each reviewer's comments.

Minor concerns:

1. Fig. 2a, Fig.3a, and 3b are not very clear. The authors should consider changing the model color, or the model display.

Reply)

In the revised manuscript, the colors of the figures have been changed to make them easier to understand for people with color blindness.

2. The assembly of RNAPII-chromatosome complexes. Why not assembled the chromatosome before loading the pol II on the template nucleosome?

Reply)

We added the H1 after the RNAPII loading, because the H1 bound to the bubble DNA region and inhibited the RNAPII loading. We described this fact in the revised manuscript (p.5, ll.100-101).

3. The authors should show the SDS-PAGE of purified proteins, and angular distributions and local resolution estimation of the cryo-EM reconstructions.

Reply)

We presented the SDS-PAGE gels of purified proteins, and the angular distributions and local resolution estimations of the cryo-EM reconstructions in Supplementary Figs. 2, 6, and 7.

4. Why did the authors choose histone H1.2? As far as I know, H1.2, which has only weak chromatin compaction ability and H1.2 does not involve direct modulation of chromatin structure.

Reply)

H1.2 is the major somatic linker histone in mammalian cells, and reportedly functions in both repression and activation of RNAPII transcription. In fact, previous genome-wide studies revealed that the genomic H1.2 binding loci negatively correlate with the RNAPII transcription activation, suggesting the repressor function of H1.2 (Cao 2013; Izzo 2013; Mayor 2015). In contrast, genetic and biochemical studies suggested that H1.2 potentially up-regulates gene expression (Brown 1996; Sancho 2008; Talasz 2009; Zheng 2010; Kim 2013). However, the effects of H1.2 on in vitro RNAPII transcription in chromatin have not been determined. Therefore, we selected H1.2 for this study. The previously reported evidence for the importance of H1.2 is described in the revised manuscript (p.4, ll.66-76).

Reviewer #2 (Remarks to the Author):

In this manuscript, the authors reported a structural study on how linker histone H1 inhibits the transcription function of RNA polymerase II (RNAPII). The system consists of a nucleosome bound to H1 with two flanking/linker DNA (92 bp and 24 bp). The longer linker DNA can form a transcription bubble with a primer RNA and is considered the entry DNA, while the other is the exit DNA. Transcription reaction shows that H1 help accumulate a ~60-nt RNA product. They then visualized the structures of RNAPII bound to the nucleosome with H1 and identified two structural forms (Form I and Form II) that have RNAPII paused at the entry DNA before H1. RNAPII moves ~2bp further on the entry DNA than Form I.

Linker histones have been proposed to inhibit transcription. However, no structure is available to help illustrate how it happens. The two structures presented in this manuscript provide the first structural snapshots of how H1 may inhibit transcription. This study should be of great interest to researchers in the fields of transcription and chromatin.

Some issues need to be clarified.

Reply)

Thank you very much for your favorable comments. According to this reviewer's

suggestions, we revised the manuscript as outlined below.

1. The authors used yeast *Komagataella pastoris* (belongs to the family of budding yeast) RNAPII and human linker histone. Budding yeast chromatin has much shorter linker DNA and no canonical H1. I suppose the authors are trying to mimic high-order systems in this study. If so, they should test if a similar reaction product is produced using mammalian RNAPII. At least the authors should discuss the rationale for choosing this hybrid system, for example, by comparing the structures of mammalian and yeast RNAPII.

Reply)

As pointed out by this reviewer, it is important to test a mammalian RNAPII in the current chromatosome transcription system. Therefore, in the revised manuscript, we prepared mammalian RNAPII from *Sus scrofa domesticus*, and performed the chromatosome transcription assay. Like the *K. pastoris* RNAPII, the *S. scrofa* RNAPII exhibited similar pausing on the entry linker DNA in the H1.2-dependent manner. Therefore, we concluded that the *K. pastoris* RNAPII pausing mechanism in the chromatosome may be conserved in mammalian RNAPII. These new data are presented in the new Fig. 1e, and the results are described in the text (p.6, ll.119-129).

2. Two bands at ~60-nt exist in the absence of H1, suggesting that the product is not determined by the specific interactions between RNAPII and H1. It needs to be pointed out.

Reply)

Thank you very much for this comment. As this reviewer pointed out, the band around 60-nt was weakly observed in the absence of H1. This suggests that the exit linker DNA may be trapped by the processing RNAPII with its DNA-binding cleft in the absence of H1, but its efficiency is low because of the flexibility of the exit linker DNA. This is consistent with the idea that H1 binding restricts the linker DNA flexibility, and enhances the RNAPII pausing rate. We discussed this point in the Discussion section of the revised manuscript (p.10, l.209-p.11, l.219).

3. The structure form I shows that the linker DNA blocks the RNAPII. It is hypothesized this is the mechanism for the inhibition. The two DNA linkers is held together by linker histone tails. It would be important to verify the hypothesis by studying the effects of removing the H1 C-terminal domain on transcription.

Reply)

Thank you very much again for this comment. As this reviewer suggested, we performed the chromosome transcription experiments with the H1.2 mutant, in which the C-terminal DNA binding region (amino acid residues 151-212) was deleted. We then found that the RNAPII pausing on the entry linker DNA was still observed, with a slightly reduced rate as compared to the full length H1. Therefore, the C-terminal DNA binding region of H1 partly contributes in the RNAPII pausing. These new data are presented in the new Fig. 1d, and the results are described in the text (p.6, ll.111-118).

4. Related to 3, this study would benefit from the inclusion of RNAPII elongation studies performed with di-nucleosomes.

Reply)

According to this reviewer's suggestion, we performed the chromosome transcription assay with di-nucleosomes, in which two nucleosomes are connected with 48 base pairs of linker DNA. The 48 base-pair linker DNA is known as an average linker length in transcriptionally active loci in human cells. We then found that the RNAPII pausing on the entry linker DNA of the upstream nucleosome was substantially enhanced by the H1 binding. Therefore, we conclude that the RNAPII pausing found in the present study may occur in the natural context of chromatin. These new results are presented in the new Fig. 4, and described in the new Results section "The downstream nucleosome does not affect the H1-dependent RNAPII pausing" (p.8, l.167-p.9, l.183).

5. If the authors believe the basic patch is critical for pausing, mutations that remove the basic patch should be made to test their hypothesis (if the experiment is doable.)

Reply)

To test the functional importance of the RNAPII basic patch, we performed a

chromosome transcription assay with the transcription elongation factor Spt4/5, which is known to mask the RNAPII basic patch. We obtained the important finding that Spt4/5 drastically alleviates the entry barrier by the chromosome formation, probably by interfering with the binding of the exit linker DNA on the RNAPII surface. This new finding provides additional evidence that the RNAPII basic patch functions to induce the H1-mediated RNAPII pausing. In the revised manuscript, we added these new data in the new Fig. 5, and discussed the in the new Results section “The transcription elongation factor, Spt4/5, alleviates the H1-mediated RNAPII pausing” (p.9, l.185-p.10, l.200).

It would also be interesting to test RNAPII mutants with basic patch mutations or deletions. However, as this reviewer noted, we need to establish a new expression system for the RNAPII mutants and their purification schemes. Therefore, it will take a long time to complete the mutational studies for RNAPII. Accordingly, we think this is an excellent project for future studies.

REVIEWERS' COMMENTS

Reviewer #1 (Remarks to the Author):

This study has been improved by the additional data and the modifications to the text. I appreciate the authors' efforts. My comments are listed below.

1. Supplementary Fig. 2e shows that nearly all of the RNA products of the cryo-EM sample were ~60nt, which is different from the results of the transcription assays (Fig1c-e). Could you provide an explanation?
2. The new data shows that the spt4/5 would alleviate the H1-mediated Pol II pausing (Fig 5b). Whether the pausing could occur in the presence of other elongation factors and/or histone chaperone (Paf1c, Spt6, FACT...)?
3. Following my previous concern, the physiological relevance of the structure still remains incompletely solved. For example, it was reported that tetranucleosome may form compact fold in the presence of histone H1, which may lead to quite different scenario. It is not a question that could be addressed by this study. However, I would suggest the review to add extended discussion and to clarify the limitation of this study. This will avoid misleading to readers.

Reviewer #2 (Remarks to the Author):

In the revised manuscript, the authors have adequately addressed my concerns.

REVIEWERS' COMMENTS

Reviewer #1 (Remarks to the Author):

This study has been improved by the additional data and the modifications to the text. I appreciate the authors' efforts. My comments are listed below.

1. Supplementary Fig. 2e shows that nearly all of the RNA products of the cryo-EM sample were ~60nt, which is different from the results of the transcription assays (Fig1c-e). Could you provide an explanation?

Reply)

Supplementary Fig. 2e shows the sample after GraFix purification for the cryo-EM analysis. In the cryo-EM sample preparation, we conducted the transcription reaction, in which a ~60-nt RNA product was predominantly accumulated in the presence of H1.2. We described these facts in the revised manuscript (p.19, ll.473-474).

2. The new data shows that the spt4/5 would alleviate the H1-mediated Pol II pausing (Fig 5b). Whether the pausing could occur in the presence of other elongation factors and/or histone chaperone (Paf1c, Spt6, FACT...)?

Reply)

According to this suggestion, we reviewed the RNAPII-nucleosome structures with the other elongation factors and FACT. We then found that Spt4/5 is the factor that directly binds to the DNA-binding cleft of RNAPII. Therefore, we speculated that Spt4/5 is the major factor for the alleviation of the chromosome transcription impediment by H1, and described this finding in the revised manuscript (p.9, ll.187-189). As the reviewer suggested, other elongation factors and/or histone chaperones could also affect chromosome transcription. This possibility is described in the Discussion (p.11, l.239-p.12, l.246).

3. Following my previous concern, the physiological relevance of the structure still remains incompletely solved. For example, it was reported that tetranucleosome may form compact fold in the presence of histone H1, which may lead to quite different scenario. It is not a question that could be addressed by this study. However, I would suggest the review to add extended discussion and to clarify the limitation of this study. This will avoid misleading to readers.

Reply)

Thank you very much for this comment. According to this suggestion, we added sentences in the last paragraph of the Discussion section (p.11, l.239-p.12, l.246) to clarify the limitations of this study and avoid misleading the readers.